# Antimicrobial Resistance and Food Animals: Influence of Livestock Environment on the Emergence and Dissemination of Antimicrobial Resistance

**DOI:** 10.3390/antibiotics9020052

**Published:** 2020-01-31

**Authors:** Nikola Vidovic, Sinisa Vidovic

**Affiliations:** 1College of Pharmacy and Nutrition, University of Saskatchewan, Saskatoon, SK S7K 4H3, Canada; nikola.vidovic@usask.ca; 2Department of Veterinary and Biomedical Sciences, College of Veterinary Medicine, University of Minnesota, Saint Paul, MN 55108, USA

**Keywords:** antimicrobial resistance, livestock, growth promotions, selective pressure, emerging infectious diseases

## Abstract

The emergence and dissemination of antimicrobial resistance among human, animal and zoonotic pathogens pose an enormous threat to human health worldwide. The use of antibiotics in human and veterinary medicine, and especially the use of large quantities of antibiotics in livestock for the purpose of growth promotion of food animals is believed to be contributing to the modern trend of the emergence and spread of bacteria with antibiotic resistant traits. To better control the emergence and spread of antimicrobial resistance several countries from Western Europe implemented a ban for antibiotic use in livestock, specifically the use of antibiotics for growth promotion of food animals. This review article summarizes the recent knowledge of molecular acquisition of antimicrobial resistance and the effects of implementation of antibiotic growth promoter bans on the spread of antimicrobial resistant bacteria in animals and humans. In this article, we also discuss the main zoonotic transmission routes of antimicrobial resistance and novel approaches designed to prevent or slow down the emergence and spread of antimicrobial resistance worldwide. Finally, we provide future perspectives associated with the control and management of the emergence and spread of antimicrobial resistant bacteria.

## 1. Introduction

The emergence and dissemination of antimicrobial resistance among human, animal, and zoonotic pathogens poses an enormous threat to people around the world [1,2]. According to a report from the World Health Organization [3], the magnitude of antimicrobial resistance in many parts of the world has reached alarming levels and clearly suggests that a ‘post antibiotic era’ may be a real possibility for the 21st century. The importance of this growing problem can be illustrated by the emergence of plasmid-mediated colistin resistance among human, animal and zoonotic pathogens, for example. Prior to the emergence of plasmid mediated colistin resistance, carbapenem-resistant *Enterobacteriaceae* that express enzymes such as KPC-2 (*Klebsiella pneumoniae* carbapenemase-2) and NDM-1 (New Delhi metallo-β-lactamase-1) already generated serious concerns around the world due to their multidrug-resistant phenotype and subsequently limited options for treatment [4]. The treatment options of any life-threatening infection caused by carbapenemase-producing Enterobacteriaceae would be severely limited, relying on only two antibiotics, tigecycline and colistin [5]. Colistin or polymixin E is a potent polycationic polypeptide that possesses both hydrophilic and lipophilic properties [6], which can effectively disorganize the outer membrane of a wide spectrum of Gram-negative bacteria. Until recently, only chromosomally mediated resistance to colistin was observed [7]. However, in 2016 the first plasmid-mediated colistin resistance was reported among several species of the Enterobacteriaceae family [8]. The authors of this report found a plasmid carrying the *mcr*-1 gene that encodes MCR-1 enzyme, of the phosphoethanolamine transferase enzyme family, which provides modification of the outer membrane and colistin resistance to bacteria that harbor this plasmid. More specifically, the expression of the *mcr*-1 gene leads to the addition of phosphoethanolamine to lipid A, which results in the reduction of affinity between colistin and lipopolysaccharides of Gram-negative bacteria. The authors of this report also observed that the plasmid carrying the *mcr*-1 gene is transmissible between several Gram-negative species including *Escherichia coli*, *Klebsiella pneumoniae,* and *Pseudomonas aeruginosa* [8]. This finding clearly indicates a likely possibility for the horizontal spread of colistin resistance to different species of *Enterobacteriaceae* and other Gram-negative species as well. Indeed, within several years of this report, the presence of the *mcr*-1 gene or its variants (e.g., from *mcr*-2 to *mcr*-8) and subsequent colistin resistance have been detected in numerous bacterial species originating from animals, humans and environments around the world [9,10,11,12].

Besides the use of colistin for human treatments, this antibiotic has been heavily used in veterinary practice, especially as an additive in feed for promoting growth of food animals in many parts of the world [13]. It has been hypothesized that the industrial-scale incorporation of colistin as a growth promoter for food animals has had a significant contribution to the emergence of colistin resistance [8]. This hypothesis is further supported by the fact that initially a high prevalence of the *mcr-1* genes was found among colistin-resistant isolates originating from animals compared to a small prevalence of the colistin resistant isolates originating from humans [8]. Examples of the emergence and spread of colistin resistance suggests that research studies designed to investigate the impact of livestock practices on the emergence of antibiotic resistance in the livestock settings and their transfer to the environment are urgently needed. This review article summarizes and presents the relevant and recent findings in the area of antimicrobial resistance and livestock production.

## 2. Mechanism of Antibiotic Resistance Development

Development of antibiotic resistance occurs via two major pathways; (i) vertically through *de novo* mutations and (ii) horizontally via the introduction of antibiotic resistance genes.

### 2.1. Vertical Acquisition of Antimocrobial Resistance

Once a population of bacteria is exposed to lethal, sublethal or even low concentrations of antibiotic, a selective pressure starts to act on the bacterial population [14]. Any mutation that will provide a full or partial resistance to a particular antibiotic will be preserved and vertically transmitted to progeny cells, further providing a survival or growth advantage (i.e., it depends on antibiotic concentration) to mutated cells compared to that of non-mutated (i.e., parental) cells. *De novo* mutations may generate bacterial resistance to a specific antibiotic exclusively, or to a class of antibiotics, when target-site gene mutation/s occur for that specific antibiotic. An example of *de novo* mutations that exclusively generate a resistance to a specific antibiotic or class of antibiotics can be found among fluoroquinolone-resistant isolates. The fluoroquinolones, a potent, broad-spectrum class of antibiotics, target bacterial type II topoisomerases, primarily DNA gyrase (GyrA and GyrB) in Gram-negative and DNA topoisomerase IV (ParC and ParE) in Gram-positive bacteria. This drug binds the GyrA or ParC enzymes at their substrate (i.e., DNA) target site [15,16], resulting in dysfunctional topoisomerase II enzymes, which are able to cleave chromosomal DNA, but unable to re-ligate the chromosome [17]. This interaction of the drug and bacterial type II topoisomerases causes a fragmentation of chromosomal DNA, which subsequently leads to cellular death [18,19]. Non-synonymous mutations that arise in the *gyrA* and *parC* genes, specifically in a short DNA region known as the quinolone resistance-determining region (QRDR), alter the substrate structure (i.e., DNA) of the GyrA or ParC [20]. These target-site amino acid substitutions lead to a reduced binding affinity of fluoroquinolone to mutated GyrA and ParC enzymes [21], further providing an exclusive resistance to fluoroquinolone class of antibiotics. 

Besides mutations that provide exclusive resistance to a specific antibiotic or class of antibiotics, most commonly, de novo mutations generate resistance to a diverse group of functionally unrelated antibiotics. Mutations that cause the overexpression of genes encoding efflux-pump proteins usually lead to the development of multidrug resistance (MDR) [22,23,24] in bacteria. For instance, the AcrAB-TolC, a member of the resistance-nodulation division (RND) family of efflux pumps, may provide MDR to Gram-negative bacteria via de novo mutations that occur in the promoter region of the AcrAB [25], mutations that occur in regulators of the AcrAB efflux pump *marA* [26], *soxS* [27], *rob* [28], and *acrR* [29] as well as mutations that affect the repressor *ramR* [24,30] of the AcrAB efflux. Efflux-pump associated mutations can not only lead to the development of MDR, but they can also cause a significant alteration in antibiotic susceptibility, developing a resistance to certain classes of antibiotics while conferring increased susceptibility to other classes of antibiotics [23]. It has been shown that de novo mutations simultaneously cause the overexpression of MDR pumps and significant down regulation of the major porins OmpF and OmpC [24], consequently leading to a reduced influx and an increased efflux of a mutated bacterial cell. In addition, decrease of porin-mediated permeability may occur during extracytoplasmic stress, via the σ^E^ regulatory loop [31], which upregulates small regulatory RNAs, MicA, RybB, and MicL that consequently antagonize synthesis of outer membrane proteins [32,33,34].

One of the adaptive strategies employed by prokaryotes during antimicrobial treatment is the development of the small colony variant (SCV), a phenotype characterized by a slow growth rate. It has been shown that the SCV-conferring mutations occur in a part of the genome that affects the electron transport and thymidine biosynthesis of bacteria [35,36,37,38,39], ultimately compromising ATP production and the electrochemical gradient across the cytoplasmic membrane [40,41]. These studies showed that the electron-transport and thymidine-biosynthesis defective SCVs mutants could be reversed to a normal (i.e., wild type) phenotype by supplementation with growth factors (i.e., menadione, hemin, and thymidine). The reversion of the SCV to a wild type phenotype by supplementation with growth factors clearly illustrates the auxotrophic nature of these SCV mutants. The acquisition of the SCV phenotype by pathogenic bacteria can have profound clinical consequences for an infected patient. The slow growth rate renders antibiotics less effective [35], promoting enhanced antimicrobial resistance among SCV bacteria, which often results in recurrent infections [42,43]. In recent years, it has become clear that SCVs play an important role in the development of chronic and antibiotic resistant infections, further contributing to an increased mortality rate of patients affected by infectious diseases [44].

### 2.2. Horizontal Acquisition of Antimocrobial Resistance

Unlike vertically acquired antimicrobial resistance, horizontal acquisition of antimicrobial resistance occurs via horizontal transfer (e.g., conjugation, transformation and transduction) of mobile antibiotic resistant genetic elements from a donor to a recipient bacterial cell. Plasmids and conjugative transposons (i.e., a small region of bacterial chromosome that encodes enzymes for its translocation) are common and important vehicles for intra and inter species or genera [45] dissemination of antimicrobial resistance. The origins of antimicrobial resistant genes are commonly identified among different environmental bacterial genera including *Kluyvera* [46], *Shewanella* [47], and *Vibrionaceae* [48]. It has been hypothesized that the potential origin of the *bla*_CTX-M_ genes, the most prevalent cause of extended-spectrum β-lactamases (ESBLs) in *Enterobacteriaceae* worldwide, was the chromosomal DNA of environmental *Kluyvera* genus [46]. It is believed that more recently the OXA-48-type carbapenem-hydrolyzing β-lactamase genes, found in *Enterobacteriaceae*, originated from the chromosome of waterborne, environmental *Shewanella* genus [47]. As illustrated by these two examples, the horizontal acquisition of antimicrobial resistance provides an effective transfer of antimicrobial resistance traits from different ecological niches to the clinically relevant species that subsequently can result in the emergence of not only multidrug resistant strains, but even pan-drug resistant strains.

Another effective vehicle for horizontal transfer of antimicrobial resistance between various bacterial species, specifically between Gram-negative bacteria, are various types of integrons [49]. These genetic elements are gene acquisition platforms that contain three genetic features necessary for the proper function of any integron, (i) *intl* gene that encodes an integron integrase, (ii) an integron-associated recombination site, *attl*, and (iii) an integron-associated promoter, Pc [50]. According to the literature, there are five classes of integrons that confer resistance to clinically relevant antibiotics [49]. The class 1 integrons are the most commonly found among clinical isolates, while classes 2 and 3 are also recovered from clinical isolates, albeit at lower frequencies compared to that for the class 1 integrons [49]. Another two classes of integrons, class 4 integrons are recovered from the human pathogen, *Vibrio cholera*, while class 5 integrons are found on the pRSV1 plasmid originated from *Alivibrio salmonicida* [49]. It has been estimated that these integrons carry about 130 different resistance gene cassettes, whose phylogeny and vast diversity indicate that these antibiotic resistance genes have been periodically captured from various genetic backgrounds [51]. It is believed that integrons play a major role in the spread of antibiotic resistance worldwide [49]. Driven by strong selective pressure, it can be expected to see an emergence and spread of integrons that confer resistance to different classes of antibiotics, antiseptics, and other harsh chemicals.

## 3. Antimicrobial Resistance Associated with Food Producing Animals

The use of antibiotics in human medicine, veterinary medicine, and agriculture has been correlated to the emergence and spread of antibiotic resistance worldwide. Especially, the use of antibiotics at industrial scale for growth promotion of food animals is believed to be a major contributor to the trend of the emergence and spread of bacteria with antibiotic resistant traits. Initially, a connection between an increase of multidrug resistance and the use of antibiotics as growth promoters was first recognized in the United Kingdom during the 1960s. This recognition was presented by the Swann Commission, which further recommended that antibiotics of human relevance should be banned for use as growth promoters for food-animals [52]. The European Union in 1999 made a step forward in the regulation of use of antibiotics in the agriculture industry by prohibiting the use of four classes of antibiotics as growth promoters, followed in 2006 by banning all growth promotion classes of antibiotics in food-producing animals [53]. The implementation of these regulations sparked debates whether the new measures designed to significantly reduce use of antibiotics in food-producing animals will have any effect on the emergence and spread of antibiotic resistance, particularly in the human population. Tang and collages [54] carried out a first systematic review followed by meta-analysis to determine the associations between measures that prohibited or reduced use of antibiotics in the food-producing animals and occurrence of antibiotic resistance in humans and animals. This research group searched numerous electronic databases and grey literature for studies that aimed to determine a relationship between any measures designed to reduce antibiotic use in food-producing animals and occurrence of antibiotic resistance in humans and animals. Based on 179 and 21 studies that described antibiotic resistance outcomes in animals and humans, respectively, this research group found a positive correlation between reducing antibiotic use in food-producing animals and the occurrence of antibiotic resistance in the studied animals. They also observed a similar correlation between reduced antibiotic use in food-animals and reduced prevalence of antibiotic resistance in humans, specifically in humans that had a direct contact with food-producing animals. A similar study carried out by Scott et al. [55], where the authors analyzed 93 studies using different animal species, antimicrobial classes, interventions, administration routes, samples and methods, showed that limiting antimicrobial use in food-animals subsequently reduces antimicrobial resistance in food-animals. These two systematic reviews, analyzing the large body of literature, undoubtedly showed that interventions designed to reduce use of antibiotics in food-producing animals indeed have a positive effect on reducing the prevalence of antibiotic resistance in both animals and humans that are in contact with food-producing animals. The implications of these measures for the general public are less certain, as a limited number of studies associated with this particular topic could be examined.

Besides this major correlation between the prohibition or reduction of antibiotic use as growth promoters and occurrence of antibiotic resistance in food-producing animals, there is another important fact to be considered during implementation of such measures (e.g., restrictions of antibiotic use in agriculture). Obviously, any implementation of measures designed to reduce use of antibiotics in food-producing animals will have multiple consequences. These measures will not only cause changes in the prevalence of antibiotic resistance but also, they will subsequently influence animal health and cost of production. McEwen et al. [56] undertook a keyword search using publicly available databases, MEDLINE and AGRICOLA, for studies that reported unintended consequences associated with national-level restrictions on antimicrobial use in food-producing animals. The search resulted in a limited number of studies (e.g., 14), exclusively from Europe. After implementation of antibiotic growth promotions (AGPs) ban in Sweden and Denmark there were initial increases of diarrhea in weanling pigs, while in other food-animal species be there was minor or no diarrhea increase observed [57,58]. It was found that in both countries, these initial problems with diarrhea incidents were successfully resolved, mainly by improving animal housing, hygiene and health management [57,58]. Similar observations were made in Norway [59] and the Netherlands [60], where initially there were reported increases in antibiotic use for the treatment or prevention of infectious disease. However, these increases in antibiotic use for infection treatment of food-animals were reduced to the previous level, prior to implementation of AGPs bans, exclusively by improvements in housing, hygiene and health managements. Although these studies came from a geographically confined area (e.g., Western Europe), the available data suggests that an increase in diarrhea incidence can be expected after an AGP ban, especially in weanling pigs. However, these initial problems are of transient nature, which do not possess any significant risk for the animals’ health and/or cost of production. Mainly through improvements in animal housing, hygiene, and health management can these problems be resolved.

## 4. Zoonotic Transmission of Antimicrobial Resistance to Humans

Transmission routes of antimicrobial resistant bacteria to humans are often extremely complex and hard to predict. In general, there are two major routes; i) direct acquisition of antimicrobial resistance through contact with the food-producing animals or human carriers and ii) indirect acquisition of antimicrobial resistance through the food chain or via exposure to niches of high antimicrobial resistance pollution (e.g., hospitals, nosocomial acquisition, manure, waste water and agriculture land).

Numerous studies, examining the transmission of antimicrobial resistant bacteria from animal to humans, reported the high prevalence rate of antimicrobial resistant bacteria among individuals that have a direct contact with animals, specifically farm workers [61,62] and veterinarians [63]. Among the first authors, Levy and colleagues [64] reported a direct transmission of multidrug resistant *E. coli* from animals to animals and also from animals to humans. This research group used *E. coli* strains that harbored R plasmid which expressed resistance to multiple families of antibiotics including chloramphenicol, tetracycline, sulphonamides and streptomycin. Once, multidrug resistant *E. coli* strains were introduced into the intestine of four chickens, and each infected chicken was caged with 50 uninfected chickens. In addition, two groups of chickens were fed on a tetracycline-supplemented feed, whereas another two groups of chickens were fed on antibiotic free feed. The authors of this study found the multidrug resistant *E. coli* strain with the test R plasmid only in chickens that were fed on a tetracycline-supplemented feed. Interestingly, over the duration of this experiment (e.g., a two-month period), R plasmid was detected in the fecal samples of human individuals that worked or lived on this particular farm. The authors of this study clearly demonstrated the importance of the zoonotic transmission route in acquisition of antimicrobial resistant bacteria. This study also highlighted the importance of the use of antibiotics as growth promoters in the spread of multidrug resistant phenotypes. Besides the existence of selective pressure, the high densities of food-producing animals, living in close quarters to one another, may also significantly contribute to the dissemination of antimicrobial resistant bacteria. A research group [65] that examined the ecology of enterohemorrhagic *E. coli* in numerous feedlot operations observed a statistically significant (*p* = 0.003) positive correlation between the density of cattle and the prevalence rate of this zoonotic pathogen. Taken together, it can be assumed that the use of antibiotics in agriculture, specifically the massive use of antibiotics for growth promotion, can lead to the emergence of antibiotic resistant bacteria. Once antibiotic resistant bacteria emerged among the food-producing animals, this phenotypic trait will be preserved by the same selective pressure and quickly spread to other animals and humans due to the high conductivity of such environments (e.g., high animal densities in the confide areas [feedlots and barns]).

Indirect acquisition of antimicrobial resistant bacteria is usually more complex compared to direct acquisition. It is known that a considerable amount of antibiotics used in agriculture, human and animal medicine reaches the environment in their active forms [66]. Subsequently, the presence of active antibiotic compounds in the environment imposes a selective pressure that may result in the emergence of antibiotic resistant phenotypes among various microbial species that naturally occupy this niche. If the antibiotic resistance phenotype emerges on a mobile genetic element, it can be horizontally transmitted to human, animal or zoonotic pathogens, which may pose a considerable public health threat. This environmental transmission route has been well documented. For instance, the origin of the *bla*_CTX-M_ genes, which are a clinically important culprit of antibiotic treatment failures, emerged from environmental Gram-negative bacteria *Kluyvera* spp. [46]. Also, the OXA-48-type carbapenem-hydrolyzing β-lactamase genes, an important cause of antimicrobial treatment failures as well, originated from the marine bacteria family *Shewanellaceae* [47]. 

Aerosols generated from areas of high antimicrobial resistance pollution represent another important vehicle that can indirectly transmit antimicrobial resistant bacteria to humans, animals and the environment in general. Madsen et al. [67] examining the presence of airborne methicillin-resistant *Staphylococcus aureus* (MRSA) in four pig farms, found the geometric mean concentrations of MRSA and *S. aureus* to be 447 colony forming units (CFU)/m3 air and 1.8 × 10^3^ CFU/m3 air, respectively. A great majority of MRSA and *S*. *aureus* were associated with particles between 7 and 12 µm in size, which can be deposited in the human upper airways, the primary, secondary and terminal bronchi and the alveoli. Interestingly, a great majority of the zoonotic transmission of MRSA isolates to humans in Europe and North America belongs to clonal complex (CC) 398, whereas CC 9 is the major livestock associated (LA)-MRSA clone in Asia [68]. For instance, a Dutch-German research group that studied a large number (*n* = 14,036) of MRSA isolates from clinical specimens of human origin in a German region characterized by a high density of livestock production, found that 18.6% of all human isolates associated with LA-MRSA CC398 based on livestock-indicator (LI) *S*. *aureus* protein A (*spa*) types [69]. Based on the same LI *spa* typing the authors of this study identified another four LA CCs among human MRSA isolates including, CC9 (0.14%), CC97 (0.01%), CC5 (1.01%) and CC30 (0.04%), indicating that CC398 is predominant among LA-MRSA in this region of Europe. In Denmark, Harrison et al. [70] using whole genome sequencing identified zoonotic transmission of novel *mecC*-MRSA ST130 between livestock and humans, indicating that LA-MRSA is highly adaptable and most likely constantly evolving. A research group from China [71], examining the effect of swine, cattle, layer and broiler farms on the spread of airborne antibiotic resistant bacteria, found that animal species are a detrimental factor in shaping total culturable and antibiotic resistant bacterial airborne communities. The importance of livestock environment on the emergence and dissemination of antimicrobial resistant phenotypes was reported by An et al. [72]. Using a well-defined population of non-typhoidal *Salmonella* (NTS) isolates associated with avian, bovine and porcine hosts, this research group found that the livestock environment had a specific (*p* < 0.005) and profound (*p* < 0.005) effect on the evolution of multidrug-resistant phenotypes among population of NTS isolates.

However, there are numerous studies that found no transmission link between zoonotic pathogens of animal and human origins. For instance, Mather et al. [73], using whole-genome sequences of a national collection of non-typhoidal *Salmonella* isolates of human and animal origins, as well as international derived isolates originated from humans and animals, found that the bacterium and its resistance genes were mainly kept within their host origins with limited transmission. Another group of authors, comparing over 430 isolates of *E. coli* (including 155 ESBL-producing isolates) isolated from livestock and retail meat with the genomes of 1,517 *E. coli* isolates associated with blood stream infections from the United Kingdom, found that these two groups of *E*. *coli* were genetically distinct populations [74]. They observed only a limited overlap in the mobile elements carrying AMR from livestock-associated and bloodstream isolates. Regarding transmission links of zoonotic pathogens and their AMR between livestock and humans, current data shows be conflicting findings and controversy, indicating that these processes are extremely complicated.

## 5. Novel Approaches Designed to Prevent or Slow Down the Emergence of Antimicrobial Resistance

There is a great need for developing alternative approaches to more effectively control the emergence and spread of antimicrobial resistance, in particular, the resistance to clinical antibiotics used in human therapy.

Over the last decade in the domain of antimicrobial research, engineered metal nanoparticles (NPs) have attracted a global attention due to their high and long-lasting microbial toxicity [75,76,77]. The antimicrobial properties of NPs are based on their high surface area-to-volume ratio, which tremendously increases the reactivity of NPs, subsequently leading to a high production of reactive oxygen species (ROS) and free metal ions. Certain metal NPs show high and broad-spectrum antimicrobial activities [78] as well as no toxicity to humans [79]. To fully exploit such properties of metal NPs, an advanced technological approach in the formulation and application of metal NPs is required. The published work of Malka and colleagues [80] showed great promise in the application of zinc-doped CuO NPs against MDR bacteria. The authors synthesized zinc-doped copper oxide NPs and subsequently deposited them on cotton fabric using ultrasound irradiation techniques. This *in situ* coating process resulted in the creation of cotton fabric with profound antimicrobial activity. A reduction of up to six logs after a 10-minute treatment was observed for both antimicrobial susceptible and MDR-strains of *E. coli* and *S. aureus*, clearly showing the outstanding antimicrobial potency of the novel Zn-doped CuO NPs against MDR strains.

The lethal effect of these novel Zn-CuO NPs is achieved by a high production of OH radicals, superoxide anions (O_2_^−^), molecular oxygen (O_2_), and most likely Zn^2+^ (Figure 1).

Once the Zn-CuO NPs were immobilized on the cotton, the antimicrobial activity of the coated fabric was stable for at least six months, denoting that the novel material has a long-lasting antimicrobial effect. Besides the use of NPs for creation of surfaces with antimicrobial properties, NPs found a promising application in conjugation with existing antibiotics. Recently, Kooti and colleagues [81] demonstrated that the antimicrobial activity of ciprofloxacin conjugated to a NP composite (e.g., graphene oxide, cobalt ferrite and silver NPs) is greatly enhanced compared to that of either NPs or ciprofloxacin alone. The authors of this study also found that this multifunctional composite could potentially be used as a drug-delivery system, mainly due its ability to gradually release drug over an extended period of time.

The use of bacteriophages has recently gained great interest, mostly due to the emergence and spread of antimicrobial resistance [82]. Bacteriophages are viruses that use bacterial cells for propagation. Only bacteriophages that undergo the lytic cycle can be used against human, animal or zoonotic pathogens, both susceptible and resistant to various antibiotics. Presently, there are two commercial *Listeria* phage products, ListShieldTM and ListexTM P100 [83] approved as food preservatives. There are several studies that tested the efficacy of this new product against *Listeria monocytogenes*. Soni and Nannapaneni [84] observed a 5-log reduction of *L. monocytogenes* after a 24 h treatment at room temperature with ListexTM P100. This product was tested also against already developed biofilms of *L. monocytogenes* and Iacumin et al. [85] observed a complete dissolution of *L*. *monocytogenes* biofilm on stainless steel wafers after applying ListexTM P100 for 24 h at 20 °C. Another group of authors that used a shorter treatment time (e.g., two hours), observed only a 2-log reduction of *L. monocytogenes* biofilm on stainless steel [86]. Another research group [87] tested reduction of *L. monocyogenes* from the surface of fresh channel catfish fillets by bacteriophage ListexTM P100. They found that the phage contact time of 30 min. was adequate to yield a greater than 1 log10 reduction in *L. monocytogenes*, whereas 15 min. contact time resulted in less than 1 log10 CFU/g reduction in *L. monocytogenes* counts on the surface of catfish fillets. Further studies designed to test the efficacy of bacteriophage treatments against various pathogens under different conditions will reveal the true potential of this promising biological intervention technology against the emergence and dissemination of antimicrobial resistant phenotypes as well as human and zoonotic pathogens.

Another antibiotic alternative is the potential use of bacteriocins, a group ribosomal synthesized peptides or proteins that show antimicrobial properties [88]. Over the last decade bacteriocins have attracted considerable attention in the area of antimicrobial research due to their distinct mode of actions. Depending on the class of bacteriocins, these antimicrobial peptides or proteins may have a bactericidal effect (e.g., causing cell death) or a bacteriostatic effect (e.g., inhibiting cell growth). Some classes of bacteriocins inhibit peptidoglycan synthesis by targeting lipid II, an integrative molecule of bacterial cell envelope [89]. Other classes of bacteriocins have binding affinity for lipid II and once attached to it they form a pore into bacterial cell envelope, further causing loss of cellular turgor, disruption of electrochemical gradient and finally, cell death [90]. Besides targeting the bacterial cell wall envelope, different classes of bacteriocins may inhibit central metabolic processes including gene expression, DNA replication, and protein synthesis [91]. Although numerous bacterial species can produce bacteriocins, most of the attention has been directed towards the lactic acid bacteria (LAB) as they are prolific producers of these antimicrobial peptides. Gomez et al. [92] examined three LAB species, *Lactococcus lactis*, *Lactobacillus sakei,* and *Lactobacillus curvatus*, against biofilms of major foodborne pathogens, *L. monocytogenes*, *E. coli* O157:H7 and *S. enterica* serovar Typhimurium. This research group observed a complete inactivation of biofilms over a 72-hour period using *L. sakei* and *L. curvatus*, while *L. lactis* caused a 6-log reduction over the same period of time. Another research group [93] used nisin, a bacteriocin approved for commercial use, against biofilm of *L. monocytogenes* over a 9-hour period and they observed a 3.5 log reduction during 48 h. There is a great diversity among bacteriocins. Alverz-Sieiro et al. [94] reported that over 230 different bacteriocins have been found to be produced by LAB. Only a small portion of these promising antimicrobial peptides have been tested.

Besides the above-mentioned alternatives, using multiple antibiotics to exploit collateral sensitivity of otherwise antibiotic resistant bacteria has shown promising results. Harrison et al. [95] showed that a significant proportion of MRSA isolates including, the epidemic USA300 lineage, can be reverted to penicillin susceptibility when used in combination with clavulanic acid, a β-lactamase inhibitor. The authors of this study concluded that although combination of penicillins and β-lactamase inhibitors most likely will not be used as a monotherapy, it can be an alternative therapeutic option for hard-to-threat infections.

## 6. Conclusions and Future Perspectives

The challenges with the emergence and spread of antibiotic resistance associated with the use of antibiotics in livestock are very complex and have multifaceted effects not only on animals, but on humans and the environment as well. It becomes clear that the use of antibiotics in livestock requires a global response, so that the measures undertaken to control the emergence and spread of antimicrobial resistance have a satisfactory effect. The first observations from countries that have already implemented bans for antibiotic use in livestock, particularly bans for growth promotion, indicate that these measures are not only sustainable, but they are profitable too. After an initial increase of diarrhea incidence, mainly in weanling pigs, improvement in animal housing, hygiene and health management resulted in reduced diarrhea incidence compared to previous levels, prior to the implementation of AGPs bans. Importantly, the recent estimates indicate that between 2010 and 2030, the global consumption of antimicrobials will increase by 67%, from 63,151 ± 1,560 tons to 105,596 tons [96]. It is believed that one third of this global increase will be related to shifting production practices in the livestock industries of middle-income countries [96]. Indeed, the most recent report, where a group of authors analyzed 901 point prevalence surveys in developing countries looking at antimicrobial resistance of pathogens isolated from animals, revealed that China and India represent the largest hotspots of resistance, with new hotspots emerging in Brazil and Kenya [97]. The same group of authors proposed that high-income countries, where antimicrobials have been used on farms since the 1950s, should support the transition to sustainable animal production in low- and middle-income countries [97]. To globally implement the WHO Global Action Plan together with the Food and Agriculture Organization (FAO) recommendations regarding the AGPs ban, it would be of a crucial importance to provide technical and financial support to developing countries, so that the entire human society can benefit fully by an AGPs ban.

Through further education of the public and professionals regarding the effects of unnecessary antibiotic use in human and veterinary medicine as well as in livestock on the emergence and spread of antibiotic resistance, it can be anticipated that the use of antibiotics will be more restricted in the future. Besides the implementation of restrictive measures for antibiotic use, it can be expected that future technological developments will significantly improve delivery and efficacy of clinical antibiotics against infectious diseases both in humans and animals. It can be assumed that the technological advances will not only improve the efficacy of the existing antibiotics, but also, they could potentially find suitable alternatives for certain types of antibiotic treatments. In the fight against the emergence and spread of antibiotic resistance, it is of crucial importance that the measures proposed by the WHO are implemented on a global scale.

## Figures and Tables

**Figure 1 antibiotics-09-00052-f001:**
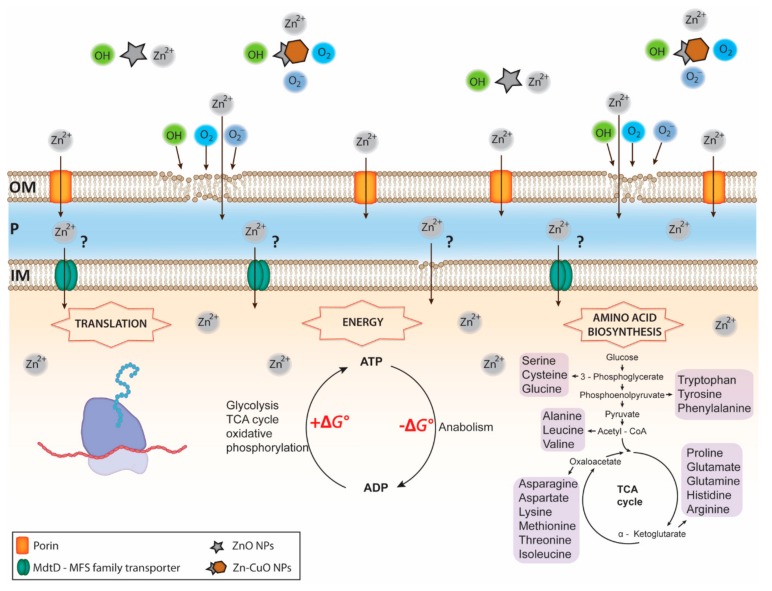
Overview of Zn NPs mediated toxicity against a Gram-negative organism. ZnO and ZN-CuO NPs release high concentrations of reactive oxygen species (ROS) and Zn^2+^, followed by Zn^2+^ entrance into the periplasmic space via porins and the outer membrane lipid peroxidation caused by ROS (i.e., OH, O_2_, and O_2_^−^). The process of lipid peroxidation leads to an increase the outer membrane permeability, resulting in an enhanced influx of Zn^2+^ into the periplasm milieu. Transport of highly concentrated Zn^2+^ across the inner membrane remains largely unknown. It can be hypothesized that Zn^2+^ enters the cytoplasm through a permeabilized inner membrane, caused by an ongoing process of lipid peroxidation and also via MFS transporters, as they can facilitate symport, antiport and uniport transfer in response to chemiosmotic ion gradients. Once penetrated, the intracellular Zn^2+^ reacts with a wide range of proteins leading to down-regulation and inhibition of enzymes and proteins involved in translation, the ATP cycle and the amino acid biosynthesis, all key metabolic processes. Disruption of these central metabolic processes leads to serious metabolic imbalances and finally to cellular death. OM, outer membrane; P, periplasmic space; IM, inner membrane.

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
