# Peer review of "Antimicrobial Resistance and Food Animals: Influence of Livestock Environment on the Emergence and Dissemination of Antimicrobial Resistance"

_antibiotics, 2020, doi:10.3390/antibiotics9020052_

Round 1

Reviewer 1 Report

REVIEW of: Antimicrobial resistance and food animals: Influence of livestock environment on the emergence and dissemination of antimicrobial resistance.

For “antibiotics”:

I must say I have no specific knowledge on this topic. I am a clinical working scientist with infection in children as my main interest.

The paper is relevant for the journal.

The topic is important.

The paper is well written and relatively easy to follow.

I have found any error.

I will recommend publication.

Author Response

Line 136 correct Enterobacteriaceae

Response # 1. Thank you for catching this mistake. It has been corrected.

Correct et al. to italics in various places.

Response # 2. “et al.” has been corrected throughout the manuscript.

Write species names in italics throughout reference list.

Response # 3. Microbial species names have been italicized throughout the manuscript. 

Is website address in reference 56 correctly formatted?

Response # 4. Reference 56, “Tang et al. Lancet Planet Health, 2017, 1, e316-e327”, does not contain a website address. It could be possible that the reviewer meant reference # 59 instead, which contains a website address. Reference # 59 has been checked and corrected accordingly. The modified website address should provide a correct source of information to the reader.     

Two different fonts are used in the reference list.

Response # 5. Thank you for catching this mistake. The reference list has been corrected.

The following references are highly relevant and the authors should consider including them:

 Van Boeckel et al. (2015) PNAS 112:5649

 Van Boeckel et al. (2019) Science

Response # 6. Thank you for pointing out these two publications. We also think that both publications are of high relevance to the topic of the current review paper. Therefore, we integrated the main findings of these two publications into our paper. 

 Check that genus names are abbreviated to first initial after first use.

Response # 7. The manuscript has been corrected. In this revised version of the manuscript, the full names of the genus have been written only on the first occasion that they have been introduced into the text. After this initial introduction, these names have been abbreviated. 

The section on alternatives is rather limited in scope and could be expanded to at least mention in brief many of the other approaches being investigated, eg. drug repurposing, vaccines, genetic engineering of resistant animals, bacteriocins and a better understanding of resistance mechanisms in order to reveal cryptic susceptibility eg. https://www.ncbi.nlm.nih.gov/pubmed/31235959.

Response # 8. Agreed. We have considerably extended this section adding some of the most attractive alternative approaches including, bacteriocins and exploiting collateral sensitivity of antibiotic-resistant pathogens. Please see the revised section.  

The role in human medicine of AMR in animals is often unclear and a few papers illustrating this would be useful for balance and to show the complexity and our lack of complete understanding. This is important given the dogma that has been allowed to develop until recently that the problem in humans arises from animals. Examples include:

 https://www.ncbi.nlm.nih.gov/pubmed/24030491

 https://www.ncbi.nlm.nih.gov/pubmed/30401778

 https://www.ncbi.nlm.nih.gov/pubmed/30670621

 https://www.ncbi.nlm.nih.gov/pubmed/30840764.

Response # 9. Agreed. We share the reviewer’s point of view. Even in our previous version of the manuscript, we indicated this complexity, lines 356-358, “The challenges with the emergence and spread of antibiotic resistance associated with the use of antibiotics in livestock are very complex and have multifaceted effects not only on animals but on humans and the environment as well.” As many of our examples in the manuscript have been pointed at the importance of livestock in human AMR, we also believe that it would be useful to include examples that show a lack of such evidence or existence of elusive evidence, so that the reader can have a sense of the complexity of this issue. Therefore, we presented and discussed findings from several articles that indicate the lack of common antimicrobial-resistant traits between pathogens of human and animal origins. Please see our revised version of the manuscript.      

As useful case study of zoonosis of resistance is this report of mecC MRSA https://www.ncbi.nlm.nih.gov/pubmed/23526809. 

Response # 10. Thank you for the link. We included findings of this interesting article in the section of livestock-associated MRSA.    

Discussion of livestock-associated MRSA particularly CC398 and CC5 would be highly relevant and useful to add.

Response # 11.  Agreed. The revised version of the manuscript contains findings from several articles indicating the importance of these clonal complexes in the emergence and spread of antimicrobial resistance from livestock to humans. 

Reviewer 2 Report

The authors deal with a task of major importance for public health, that of antibiotic resistance and some of the issues that may lead to its generation and transmission along the food chain, among animals and from the environment to the humans. In general the review is well written and describes well the major aspects that are involved in the generation and establishment of antibiotic resistance. Although the references are adequate and properly selected, I would like to see some of those that go back to more than two decades to be replaced by some earliest ones. Otherwise, I have no other major comment to make.

Author Response

(The authors gave the same response as above.)
